# Rice Husk Hydrolytic Lignin Transformation in Carbonization Process

**DOI:** 10.3390/molecules24173075

**Published:** 2019-08-24

**Authors:** Svetlana Yefremova, Abdurassul Zharmenov, Yurij Sukharnikov, Lara Bunchuk, Askhat Kablanbekov, Kuanish Anarbekov, Tetiana Kulik, Alina Nikolaichuk, Borys Palianytsia

**Affiliations:** 1National Center on Complex Processing of Mineral Raw Material of the Republic of Kazakhstan RSE, Almaty 050036, Kazakhstan; 2Chuiko Institute of Surface Chemistry National Academy of Sciences of Ukraine, Kiyv 03164, Ukraine

**Keywords:** hydrolytic lignin, rice husk, pyrolysis, carbonization, carbon structure, IR, TPD-MS, XRD, TEM, EPR

## Abstract

Lignin processing products have an extensive using range. Because products properties depend on lignin precursor quality it was interesting to study lignin isolated from rice husk being a large tonnage waste of rice production and its structural transformations during carbonization. Lignin isolated by the thermal hydrolysis method with H_2_SO_4_ 1 wt % solution and its carbonized products prepared under different carbonization conditions were characterized using elemental analysis, IR, TPD-MS, XRD, TEM, and EPR. It was shown lignin degradation takes place over the wide (220–520 °C) temperature range. Silica presenting in lignin affects the thermal destruction of this polymer. Due to the strong chemical bond with phenolic hydroxylic group it decreases an evaporation of volatile compounds and as a result increases the temperature range of the lignin degradation. Rice husk hydrolytic lignin transformations during carbonization occur with generation of free radicals. Their concentration is decreased after condensation of aromatic rings with carbon polycycles formation, i.e., the graphite-like structure. Quantity and X-ray diffraction characteristics of the graphite-like phase depend on carbonization conditions. Morphology of the lignin-based carbonized products is represented by carbon fibers, carbon and silica nanoparticles, and together with another structure characteristics provides prospective performance properties of lignin-based end products.

## 1. Introduction

Lignin is one of the largest renewable raw materials of aromatic nature [1]. Lignin processing has been a theme of current interest over the world especially in the last years [2,3,4]. Unfortunately, in some countries [5] including of Kazakhstan lignin continues to be considered as a constituent part of biomass, which has low commercial value and, moreover, creates environmental problems due to accumulation in large quantities. However, a structure of lignin predetermines the possibility to obtain various chemical compounds and materials during its processing. The application sphere of these products is very diverse [2,3,6]. It is well-known [4,7,8,9,10] that properties of lignin-based products depend largely on lignin quality. Lignin quality in turn depends on the plant feedstock and lignin isolation process used. In addition, different technological parameters of the same method of lignin production also affect its structure [8] (p. 2) and can be the reason for changes in properties of end lignin-based products. Various lignin isolation methods have been known since the 1950s. These methods can be divided into two groups: (1) methods based on the translation of carbohydrates in solution and obtaining lignin in the form of insoluble residue; (2) methods based on the dissolution of lignin followed by its precipitation from the resulting solutions. Classic and modified methods of wood/biomass treatment using ball milling, organic solvents, acids, and others have been presented widely in different years [9,11]. Each of them has advantages and disadvantages. For example, ball milling of wood followed by lignin extraction using neutral solvents at room temperature is considered to be most advanced method to produce unmodified lignin. However, the degree of lignin structure change and the degree of the isolated lignin representativeness from its total content in wood remain controversial. Organosolv treatment of wood depending on solvent type, temperature, processing time, and kind of wood can produce original native lignin but its yield is low (2–3 wt %). An important method of lignin isolation is treatment of wood by concentrated mineral acids (H_2_SO_4_, HCl) at room temperature, because this method allows detecting a quantitative content of lignin. Thermal hydrolysis by a dilute solution of sulfuric acid at 170–190 °C is recommended to avoid strong isolated lignin change [11]. The benefit of this way is possibility to produce lignin as an insoluble residue during one stage. Therefore, it is necessary to select the most suitable method of lignin isolation depending on specific use.

A considerable direction of lignin raw materials processing is the production of carbon materials, such as: fibers, sorbents, and fillers [10,12,13]. The reason is a desire of manufacturers of carbon materials to reduce their cost because their main production cost is associated with the price of precursor and carbonization process [12] (p. 284). However, there is opinion that lignin being a natural adsorbent is relatively non-reactive in the activated carbon production and a lack of detailed researches in this direction [14] encourages finding out more and filling gaps in the field of lignin carbonization process.

In this context, the aim of current research was to study structural transformations of isolated from rice husk hydrolytic lignin during the carbonization process. Kazakhstan, like many other Asian countries is engaged in rice cultivation. Rice processing leads to the formation of large-tonnage waste of rice husks. Rice husk makes up 20% of the mass of unshelled rice. While the scale of rice production in Kazakhstan is significantly less compared to China or India for example, the problem of rice husk processing is relevant as in these countries. The number of methods of rice husk processing developed is huge [15]. However, rice husk up to now has not found an effective industrial application, because of economic and ecological factors, and is sent to dumps. This requires the rejection and removal from the national economic turnover of more and more areas of land, as rice husk is not subject to decay because it contains large amount of silica [16] (p. 263). Searching for the most efficient method of rice husk processing, we concluded that main components of rice husk—which are cellulose and lignin—impact structure and properties of produced carbon materials differently because of their completely different structures [17]. This fact was the basis for a solution about feasibility to isolate in some cases the components from rice husk in order to use them to produce materials with a required structure and properties. In the frame of this study the lignin from rice husks was isolated using a thermal hydrolysis method and carbonized in an atmosphere of outgoing steam gas and under vacuum at different temperatures. Some investigation methods—such as infrared spectroscopy (IR), temperature-programmed desorption mass spectrometry (TPD-MS), X-ray diffraction analysis (XRD), transmission electron microscopy (TEM), and electron paramagnetic resonance (EPR)—were used in order to ensure a comparative study of initial lignin and obtained lignin-based carbon materials. The lignin structural transformations during its carbonization were established as well as morphology and structure of lignin carbonization products explaining their fine performance properties, especially when they were used as fillers for composite materials.

## 2. Results

### 2.1. Yield and Composition of Hydrolytic Lignin and Products of its Carbonization

The hydrolytic lignin yield was 37 wt % at a solid-to-liquid ratio of 1:8 and hydrolysis duration for 1 h and in other case it was 34 wt % at a solid-to-liquid ratio of 1:20 and hydrolysis duration for both 1 h and 2 h during its isolation process from rice husk with 1 wt % of sulfuric acid solution. It was determined that a hydrolytic lignin sample (L) obtained consists of 58% lignin, 10% polysaccharides, and 32% silicon dioxide (there is more than 15% of silica in rice husk composition used as a precursor).

The yield of hydrolytic lignin carbonization products varied depending on carbonization conditions. It was averaging more than 75 wt % when carbonization was carried out under stationary conditions in the atmosphere of outgoing steam gas (OSG) under atmospheric pressure in a temperature range from 300 to 1000 °C for 30 min. At the same time, yields of carbonization products decrease with an increase in the pyrolysis temperatures due to greater primary decomposition of precursor at higher temperatures as well as secondary decomposition of the carbon substance formed [18]. Realization of the carbonization process under vacuum (with a residual pressure of 10–20 kPa), under otherwise equal conditions, was the reason for decrease of carbonization products yields by an average to 60 wt % due to accelerated removal of gaseous products from reaction zone and enhancing of destruction processes of the organic part of raw material. In general, the high yields of carbonized products are explained by the presence of a large quantity of silicon dioxide in the hydrolytic lignin.

The elemental quantitative composition of the rice husk hydrolytic lignin carbonization products is represented mainly by carbon combined with silicon dioxide, and it varies depending on the carbonization conditions as shown in Table 1. As can be seen in the Table 1, there is observed a domination of carbon in composition of carbonization products obtained in the atmosphere of outgoing steam gases. The mineral residue, represented mainly by silicon dioxide, is a predominant phase of carbonization products obtained under vacuum.

### 2.2. Investigation of Rice Husk Hydrolytic Lignin Structure and its Carbonization Products by Infrared Spectroscopy

In spectrum of rice husk hydrolytic lignin, the specific set of bands was fixed (Figure 1). These bands are caused by vibrations of: hydroxyl groups at 3416 cm^−1^; aliphatic groups at 2925 cm^−1^; conjugated aldehyde and ketone groups as well as unconjugated carbonyl and carboxylic groups in the range of 1700 cm^−1^; C=C aromatic bonds at 1600 cm^−1^ and 1512 cm^−1^ (the band is characteristic for lignin guaiacyl units); methylene CH_2_-groups at 1450 cm^−1^ [19]. The absorption bands at 468, 792, and 1100 cm^−1^ are conditioned by the presence of amorphous silicon dioxide, although the bands in the region of 1100–1200 cm^−1^ might be caused by vibrations of C–H in syringyl (at 1113 cm^−1^) and guaiacyl (at 1031 cm^−1^) units and C−C, C−O, C=O bonds (at 1214 cm^−1^) in lignin guaiacyl units [12] (p. 292).

A transmittance curve as can be seen in Figure 1 in IR spectrum of hydrolytic lignin carbonized at 500 °C (L-500) decreases in the region from low to high frequencies. The band at 3416 cm^−1^ is blurred, and the band at 2925 cm^−1^ disappeared at all (Figure 1). The absorption bands at 1696 and 1512 cm^−1^ disappeared as well, the band at 1450 cm^-1^ is still detected as a trace. The band at 1604 cm^−1^ shifted to a low-frequency region (to 1590 cm^−1^), at that point its broadening occurred. The intensity of the absorption band at 464 cm^−1^ decreased slightly, while the bands intensities at 800 and 1104 cm^−1^ showed opposite trends with the increasing slightly. IR spectra of solid residues of hydrolytic lignin thermal decomposition at temperatures above 500 °C (L-650; L-800) are less scattered due to increased background, but the absorption bands of silicon dioxide are recorded in the low-frequency region (up to 1100 cm^−1^).

### 2.3. Study of Hydrolytic Lignin Transformation in the Carbonization Process by Temperature-Programmed Desorption Mass Spectrometry Method

The peaks on (P–T) curves of lignin-based carbonized products shift toward high-temperature area (Figure 2). The higher pyrolysis temperature of plant precursor is, the more shift of the peaks on the appropriate (P–T) curve toward high-temperature area is. For example, L-400 has maximum at 550 °C on the (P–T) curve unlike L-600 which loses volatile compounds at temperature as higher as 550–750 °C. At the same time the pressure of volatile compounds is lower in the case of L-600 that contains their less quantity (6.1% instead of 15.6% for L-400). The pressure of volatile pyrolysis products in comparison with initial L decreases in case of L-400 approximately by 1.5 times and almost by an order in case of L-600.

The analysis of mass spectra of pyrolysis products in a wide temperature range has shown (Figure 3) that thermal decomposition of lignin sample L begins above 150 °C due to decomposition of present polysaccharide residues. This is evidenced by the presence of fragment ion with *m*/*z* 60 (HOCHCHOH^+^), which is the most intense marker ion in mass spectra of pyrolysis products of carbohydrates [20,21].

As can be seen in Figure 3, pyrolysis of lignin takes place over the range from ~220 °C to ~520 °C. The main degradation products of lignin macromolecule are phenolic compounds, such as: phenol (*m*/*z* = 94, T_max_ = 355 °C), pyrocatechol (*m*/*z* = 110, T_max_ = 384 °C), guaiacol (*m*/*z* = 124, T_max_ = 340 °C), syringol (*m*/*z* = 154, T_max_ = 355 °C), *o*-cresol or *p*-cresol (*m*/*z* = 107, 108, T_max_ = 370 °C). Their formation is caused by the thermal transformations of the corresponding phenylpropanoid blocks of lignin: paracoumaryl-, coniferyl-, and sinapyl-blocks. Phenols are characterized by the different temperatures of maximum desorption rate T_max_ (Table 2). Their TPD peaks are broadened due to formation of phenol molecule which can be in progress as a result of thermal transformations of various phenylpropanoid fragments and breakdown of various types of bonds. In other words, TPD peaks of phenols are a superposition of several TPD reactions. The kinetic parameters of the formation of only pyrocatechol and methylguaiacol were calculated in this study (Table 2). Based on calculated negative values of activation entropy, it can be said that the formation processes of these compounds run through highly ordered cyclic transition states.

The formation of the following vinyl derivatives in small amounts during the lignin pyrolysis process was established (Figure 3, Table 2): 4-vinylphenol (*m*/*z* = 120), 4-vinylpyrocatechol (*m*/*z* = 136), 4-vinylguaiacol (*m*/*z* = 150), 4-vinyl-methylguaiacol (*m*/*z* = 164). Calculated kinetic parameters indicate that these processes are also characterized by negative values of activation entropy. The formation of vinyl derivatives can be caused by the presence of silicon dioxide nanoparticles in lignin composition [22,23,24,25]. It is known that the accumulation of silica in plants occurs when ferulic acid is involved [26]. There is a certain amount of cinnamic acids fragments conjugated with silica in lignin of rice husk, which can be revealed by the presence of an absorption band at 1696 cm^−1^ (Figure 1) caused by stretching vibrations ν_C = O_ [23,25,26,27]. Cinnamic acids can interact with silica surface through both carboxylic and phenolic functional groups [23,24]. This complex biochemical process proceeds as a result of the interaction of arabinoxylan-ferulic acid complexes with hydroxylic groups of silicic acid Si(OH)_4_ [23,24,25,26].

There is a high-temperature stage of hydrocarbons desorption as well, including benzene, phenylacetylene, naphthalene, 9H-fluorene, anthracene, or phenanthrene (Figure 3 and Figure 4 , Table 2) in the process of rice husk hydrolytic lignin pyrolysis. Certainly, this is a natural result of the condensed aromatic and graphite-like structures formation. However, it should be noted the role of present silica, which is capable to form a strong chemical bond with phenolic hydroxyl group PhO-Si ≡ [28,29]. Chemical graft of biomolecule to silica surface prevents such complexes from desorption at lower temperatures. Therefore, thermal transformation of Ph-O-Si ≡ graft-complexes carries out with the formation of polymer aromatic coating on the surface of silica, and destruction of this surface coating takes place at temperatures above 500 °C (Figure 4, Table 2) [23,24].

### 2.4. X-ray Phase Composition and X-ray Diffraction Characteristics of Rice Husk Hydrolytic Lignin and Its Carbonization Products

Cellulose presence (up to 20%) was found in the sample of rice husk hydrolytic lignin using X-ray diffraction analysis. It was identified from reflections with interplanar distances *d* = 0.56 nm and 0.394 nm (crystalline cellulose phase marked as C_cr_ in Figure 5a). This fact actually confirmed the results of chemical analysis and TPD-MS listed above. In order to increase chemical purity and uniformity of hydrolytic lignin (amorphous phase marked as A in Figure 5a), it was subjected to treatment with 72 wt % of sulfuric acid. The X-ray diffraction pattern of the purified hydrolytic lignin exhibited a wide halo with a maximum at θ = 11° with *d* = 0.4 nm (Figure 5b). This shape is typical for amorphous organic substances containing hydrocarbon phases [30,31]. According to nature and angular position of (002) band, the lignin is comparable to main hydrocarbon component of carbonaceous materials previously known as γ-phase (*d* = 0.47 nm), and later called ‘boghed-like’ (polynaphtenic). The polynaphtenic phase has a clathrate structure consisting of containing alkane chains naphthenic cycles both condensed and separated by methylene bridges [32] and lignin also consists of structural units of C_6_-C_3_ oxygen derivatives of phenyl propane.

X-ray diffraction patterns of hydrolytic lignin carbonization products exhibited a wide halo in the region of θ from 5 to 16° (Figure 5a). It is typical for carbon materials which are multicomponent systems. X-ray diffraction patterns obtained were decomposed into components by an iterative method using approximation based on experimental profiles of reflections for a number of standard substances [13] (p. 78). This fact allowed to reveal the presence of carbon (graphite-like, G) and two hydrocarbon (polynaphtenic, N; and unidentified structure, probably oxygen-containing, H) phases, characteristics and amount of which are given in [13] (p. 81). Only two phases (N and H) were fixed in hydrolytic lignin carbonized at 400 °C (L-400). The graphite-like phase appearance and mutual transformations of N and H are observed with rising of carbonization temperature (Table 3) [13] (p. 81). Hydrolytic lignin carbonization conditions affect X-ray-diffraction characteristics of G in carbonizations products prepared as well. Their interplanar distances (*d_002_*) values remain the same in the temperature range of 500–650 °C, but with temperature increasing up to 800 °C they decrease and then stabilize again. Based on this and taking into account the rise of crystallites stack thickness *L_c_* (Table 3), it can be said that mainly structure transformations take place as a result of an intrablock interlayer orientation. An increase in the interlayer ordering proceeds to 800 °C. Condensation processes in carbon layers begins to increase at higher temperature as indicated by the appearance of weak (100) bands in the corresponding X-ray diffraction patterns in the θ region of 21–22 °. However, due to these bands blurring, it was impossible to calculate the average diameter of the planar fragments of crystallites (*L_a_*), i.e., size of the layers of cyclically polymerized carbon.

### 2.5. Study of the Morphology of Hydrolytic Lignin Particles and its Carbonization Products by Transmission Electron Microscopy

It is known that morphology and particles size of lignin depend on conditions of its production [6] (p. 8), [10]. Morphology of hydrolytic lignin isolated from rice husk using a solution of sulfuric acid is represented by fiber networks (Figure 6a) and dendritic formations (Figure 6b) formed by particles with rounded shape and nanometer size (Figure 6c). Nanoparticles of silicon-containing substance were fixed using microdiffraction pattern which is represented by point reflexes with *d* = 0.449 nm (Figure 6d).

Morphology of carbon particles obtained during the hydrolytic lignin carbonization in general is the same as morphology of precursor (Figure 7). There are coils of disorderly interwoven carbon fibers or tapes (Figure 7a,b), spherical particles (Figure 7b), as well as dendritic and large isomorphic formations (Figure 7c). Apparent size of fibers in diameter varies from 20–30 nm (Figure 7a) to 80–100 nm (Figure 7b). Size of spherical particles reaches 230–260 nm (Figure 7b). Dendritic structures are formed by rounded particles which are 30–50 nm in size, large formations consist of grains with size of 3–5 nm (Figure 7c). According to the microdiffraction patterns, matter of tested samples is mainly electron-amorphous. The rounded particles presented in Figure 7b,c do not show structural ordering. Their microdiffraction patterns are represented by diffuse rings with a set of reflexes 0.371; 0.198 nm (Figure 7d). These data are similar to those of X-ray diffraction analysis.

### 2.6. Investigation of Structural Changes of Hydrolytic Lignin During Carbonization by EPR Spectroscopy Method

It was established using the EPR spectroscopy method that rice husk hydrolytic lignin shows an EPR signal at even room temperature as it can be seen in Figure 8a (spectral parameters are presented in Table 4). It might be caused by the formation of free radicals during chemical treatment of rice husk to isolate lignin, the presence of multiple bonds as well as long-lived free radicals of quinone or phenoxy-type. Concentration of paramagnetic centers (PMCs) during carbonization of hydrolytic lignin at low temperatures (up to 300 °C, Figure 8b, Table 4) decreases probably due to the thermal recombination of initial stable radicals [33], and then it grows again reaching a maximum value (3.1·10^17^ spin g^−1^) at 550 °C (Figure 8d, Table 4). The value of this parameter falls at higher temperature (650 °C, Figure 8e, Table 4) as a result of a condensation of carbon rings forming graphite-like structure.

The EPR line width tends to decline in general with an increase in the hydrolytic lignin carbonization temperature (Figure 8a,b,d,e). A decrease in the EPR line width with increasing concentration of paramagnetic centers can be explained by the strengthening of exchange interactions in the system of spins of free radical states. A decrease in the EPR line width with a drop in the concentration of paramagnetic centers can be caused by a decline in the dipole–dipole interaction and the appearance of delocalized π-electrons in graphite-like structure. However, an EPR line width broadening is observed at 450 °C (Figure 8c). Probably, the paramagnetic centers of two types (free radicals and clusters resulting from the closure of free radicals) are formed at this temperature. They have similar values of g-factor, and the superposition of their signals leads to the broadening of EPR line.

## 3. Discussion

X-ray diffraction analysis results have shown that the formation of graphite-like component takes place during carbonization of rice husk hydrolytic lignin. Moreover, the interplanar distance *d_002_* decreases while average thickness (height) of crystallite stack *L_c_* increases with carbonization temperature rising. This fact indicates that the process of graphitization grows with an increase in the carbonization temperature. The comparative analysis data of hydrolytic lignin carbonization products and initial hydrolytic lignin carried out by transmission electron microscopy have shown that supramolecular structure of carbon-containing materials, consisting of many small grains, repeats supramolecular structure of precursor, i.e., hydrolytic lignin. This result confirms a thesis noted in a paper [12] (p. 293) that the quality of initial lignin precursor impacts the quality of produced carbon fibers.

Comparing the regularity of concentration of paramagnetic centers changes depending on the hydrolytic lignin carbonization temperature rise and results of IR spectroscopy, TPD-MS and XRD, it can be stated that an increase in the PMCs concentration up to ~550 °C is caused by the free radical states rising as a result of splitting of energetically weak bonds and removal of easily mobile groups. The construction of condensed carbon rings forming a graphite-like structure leads to the paramagnetic centers concentration fall. This may be caused by the formation of nanostructures as well [34]. Consequently, the EPR line width loss observed at an increase in the hydrolytic lignin carbonization temperature (Table 4) is explained by the rise of exchange interactions in spins system of free radical states in the case of paramagnetic centers concentration growth and by a degradation of dipole–dipole interaction and appearance of delocalized π-electrons in graphite-like structures in the case of paramagnetic centers concentration decrease. A gradual decline in g-factor values which are closed to g-factor of free electron (g = 2.0023) in graphite structures is also observed in the rice husk hydrolytic lignin carbonization temperature range under consideration.

It is interesting to compare the dynamics of structural characteristics changes of rice husk hydrolytic lignin-based and rice husk-based carbonized products. Hydrolytic lignin compare with rice husk is more thermostable. High yield of rice husk hydrolytic lignin carbonization products compare with rice husk-based products obtained under the same conditions is evidence of it [35]. By the way, it explains why hydrolytic lignin has lower reactivity than rice husk in the carbonization process considered [13]. Lignin is relatively thermostable plant raw material. A content of graphite-like component can be also considered as an indicator of thermal stability. The content of graphite-like component with interplanar distance 0.375 nm in the composition of hydrolytic lignin carbonization product at 1000 °C is 62% (Table 3) while it reaches 100% in composition of identical product of rice husk carbonization [13] (p. 81). At the same time, there is appeared clear regularity, the less a quantity of graphite-like component in compared carbonized products, the more a value of specific surface area of corresponding samples [36]. However, an increase in the paramagnetic center concentration is a more demonstrative characteristic of thermal stability of hydrolytic lignin. This index grows up to 550 °C during hydrolytic lignin carbonization (Table 4) and up to 450 °C during rice husk carbonization [37]. A condensation of carbon rings with a formation of graphite-like structure proceeds at higher temperatures. This process develops more intensively in the range of temperatures up to 1000 °C in case of rice husk. On the other hand, products of hydrolytic lignin carbonization at temperatures above 1000 °C are characterized by higher structural order. For example, a graphite-like component with *d_002_* of 0.336 nm is a predominant phase (75%) in lignin-based material obtained at 1650 °C while the content of the same phase in analogous rice husk carbonization product does not exceed 53% [36] (p. 154).

A feature of hydrolytic lignin carbonization products is the relative uniformity of the shape (carbon fibers and dendritic formations formed by spherical grains) and particle size (at nano level). Due to the presence of cellulose large (micron size) and shapeless formations are in rice husk carbonization products in addition to nano-particles caused by the lignin presence [17] (p. 1114). Morphology of carbon-containing materials, determining their operational characteristics, is an important indicator for their using in practice, in particular as fillers for composite materials. It is known [38] that rice husk hydrolytic lignin is used as an ingredient of carbon self-lubricating material, which is applied in mechanical seals and plain bearings in order to ensure the reliable performance under varying conditions, for example, boundary friction, and in mediums containing of abrasive particles, for example, in water and chemical solutions pumps, in centrifugal pump units of ship installations, as well as the equipment of chemical and petrochemical industry. It has been established that rice husk carbonization products can be successfully used in the production of structural friction and antifriction materials. It was shown in previous papers [13] (p. 80), [35] (p. 208) that in comparison with technical carbon, fillers produced using rice husk and its derivatives (lignin and cellulose) considerably improve technological characteristics of rubber mixes and finished vulcanizates, such as in elasticity, adhesion, and strength properties. The reason is firstly, presence of a carbon and silica mix and secondly, a structure of carbon matrix. The results of current study confirm this thesis and help to explain the higher activity of the hydrolytic lignin carbonization product as a filler of composite materials in comparison with similar product produced from rice husk. The advantage of hydrolytic lignin-based filler in addition to X-ray phase composition and X-ray diffraction characteristics of carbon structure is shape and size of carbon particles. Carbon particles of hydrolytic lignin carbonization products correspond to silicon dioxide particles presenting in both hydrolytic lignin and rice husk carbonization products in shape and size. Therefore, hydrolytic lignin-based filler acts like homogeneous nanomaterial as a result it improves performance properties of filled products much more than rice husk-based filler.

To conclude, structural transformations of rice husk hydrolytic lignin in the process of carbonization proceed through the stage of free radical formation followed by hexagonal networks formation of cyclically polymerized carbon. Decomposition rate of hydrolytic lignin, the concentration of free radicals forming during this process, as well as a quantity and X-ray diffraction characteristics of the graphite-like component depend on the carbonization conditions. These indicators, together with morphology, predetermine properties and respectively the activity of lignin-based carbon products. Therefore, it is necessary to continue study to find out change of properties of rise husk hydrolytic lignin based carbon materials (for example, sorbents, fillers) depending on their structure characteristics found in current work to recommend their production conditions for specific uses.

## 4. Materials and Methods

### 4.1. Materials

#### 4.1.1. Rice Husk Characteristics and Pretreatment

Rice husk (RH) obtained from an agricultural enterprise in Kyzylorda region was used. Rice husk was washed with distilled water at room temperature at a solid-to-liquid (S/L) ratio of 1:5 for 10 min. Then it was separated from water and dried firstly at room temperature for 15 h and secondly at 105 °C until the mass remained constant.

Rice husk processed in this way consists of (wt %, dry basis): cellulose (32.8), hemicellulose (17.2), lignin (25.5), extractives, soluble in an alcohol–benzene mixture (1.8) and hot water (6.2), silicon dioxide (15.5%), and other mineral substances (Na—0.039; K—0.355; Ca—0.05; Mg—0.065; Fe—0.034; Al < 0.1) [16] (p. 264).

It was used as a precursor to produce hydrolytic lignin (L).

#### 4.1.2. Hydrolytic Lignin Preparation and Carbonization Processes

Hydrolytic lignin was isolated from rice husk using the method known as thermal hydrolysis with H_2_SO_4_ 1 wt % solution at 200 °C at a S/L ratio of 1:8 and 1:20 for 1 h and 2 h. After cooling the autoclave, the solid residue was filtered through a Buchner funnel, and washed with hot distilled water to neutrality. The residue, which had a brown color, was dried until the mass remained constant.

Prepared hydrolytic lignin sample was purified with H_2_SO_4_ 72 wt % solution according to Komarov method [39].

Hydrolytic lignin sample of 100 g were heated in a furnace at a constant heating rate 15 °C min^−1^ at a temperature of 300–1000 °C for 30 min in atmosphere of outgoing steam gas and under vacuum (10–20 kPa). The prepared carbonized products depending on carbonization temperature marked as L-300, L-400, L-450, L-500, L-550, L-600, L-650, L-800, and L-1000 were cooled without air to room temperature.

### 4.2. Analytical Methods

#### 4.2.1. Elemental Analysis

Elemental analysis was performed on the Khimlaborpribor facility (Klin, Russia).

#### 4.2.2. Infrared Spectroscopy

IR spectra were obtained on a Specord M80 spectrophotometer (Carl Zeiss, Jena, Germany). The samples were grinded, mixed and ground with KBr to form pellets with the concentration of tested substance of 1.0–3 wt %. The spectra were recorded between 4000 and 400 cm^−1^. The IR spectra presented are the average of three measurements.

#### 4.2.3. Temperature-Programmed Desorption Mass Spectrometry

Temperature-Programmed Desorption Mass Spectrometry (TPD-MS) was performed on MKh-7304A monopole mass spectrometer (Electron, Sumy, Ukraine) with electron impact ionization, adapted for thermodesorption measurements. A typical test comprised placing a 10 mg sample on the bottom of a molybdenum-quartz ampoule, evacuating to approximately 5 × 10 ^−5^ Pa at room temperature and then heating at 0.15 °C s ^−1^ to approximately 750–800 °C. Volatile pyrolysis products were passed through a high-vacuum valve (5.4 mm in diameter) into the ionization chamber of the mass spectrometer where they were ionized and fragmented by electron impact. After mass separation in the mass analyzer, ion current due to desorption and pyrolysis was amplified with a VEU-6 secondary-electron multiplier (“Gran” Federal State Unitary Enterprise, Vladikavkaz, Russia). Mass spectra and (P–T) curves (where ‘P’ is the pressure of the volatile pyrolysis products, and ‘T’ is the temperature of the samples) were recorded and analyzed using a computer-based data acquisition and processing setup. The mass spectra were recorded within 1 to 210 amu. During an each TPD-MS experiment, approximately 240 mass spectra were recorded and averaged. During a thermodesorption experiment, samples were heated slowly while keeping a high rate of evacuation of the volatile pyrolysis products. Diffusion effects can thus be neglected, and an intensity of ion current can be considered proportional to a desorption rate [40].

#### 4.2.4. X-ray Diffraction Analysis

X-ray diffraction patterns of the investigated materials were determined using a DRON-2 computerized diffractometer with modernized collimation using filtered Cu K_α_ radiation in reflected rays in a 17 × 17 × 1.5 mm^3^ metal cell. Measurements were recorded in the θ range of 4° to 25°. The concentrations of the components were found by quantitative X-ray phase analysis with an error of ±5% of the concentration of each phase [13] (p. 78).

#### 4.2.5. Transmission Electron Microscopy

Transmission electron microscopy (TEM) was performed in a JEM 100 CX transmission electron microscope (JEOL, Tokyo, Japan) at an accelerating voltage of 100 kV. Samples powdered using an agate mortar were put on carbon coated copper grids and fixed in the microscope object holder.

#### 4.2.6. EPR-Spectroscopy

Study of samples by EPR-spectroscopy method were made using a modernized homodyne EPR IRES-1001-2M spectrometer of homodyne type operating in wavelength region (3 cm). The measurements were performed at room temperature on a sweep of a magnetic field of 120 Oe with a microwave power of 16 mW, and a modulation amplitude of 1 Oe. These conditions of spectra recording were chosen as optimal because microwave power value eliminated the effects of EPR saturation and magnetic modulation amplitude eliminated broadening of EPR line. EPR spectra were recorded between 3 and 4 components of the reference sample which were ions Mn^2+^ in MgO. G-factor and an EPR line width of tested samples were determined using known reference sample parameters. The concentration of free radical states of tested samples was calculated by comparison of areas of their spectra and calibrated reference sample (third line of EPR spectrum ions Mn^2+^ in MgO).

## Figures and Tables

**Figure 1 molecules-24-03075-f001:**
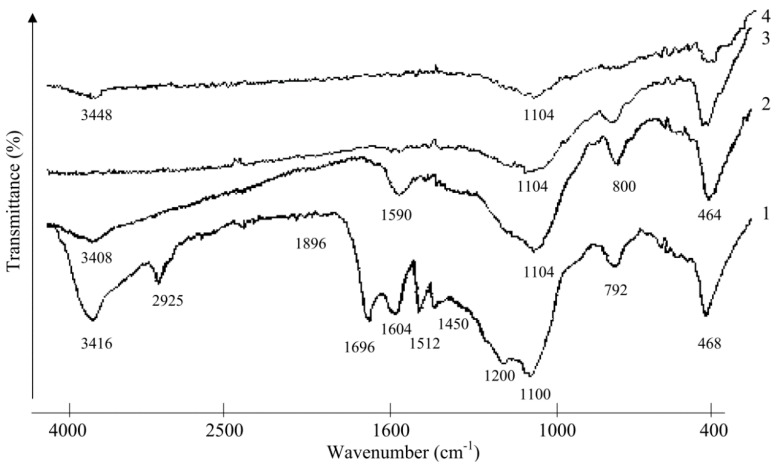
IR spectra of hydrolytic lignin and its carbonization products: 1—L (3 mg); 2—L-500 (3 mg); 3—L-650 (3 mg); 4—L-800 (1 mg).

**Figure 2 molecules-24-03075-f002:**
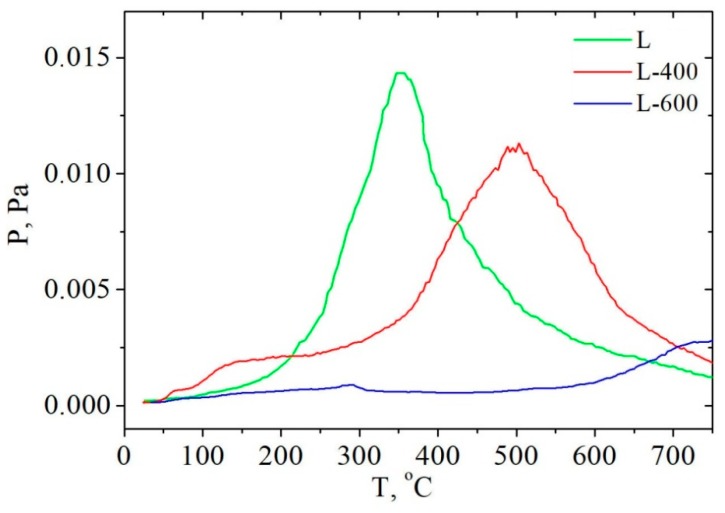
Temperature-pressure (P–T) curves of rice husk hydrolytic lignin and its carbonization products pyrolysis.

**Figure 3 molecules-24-03075-f003:**
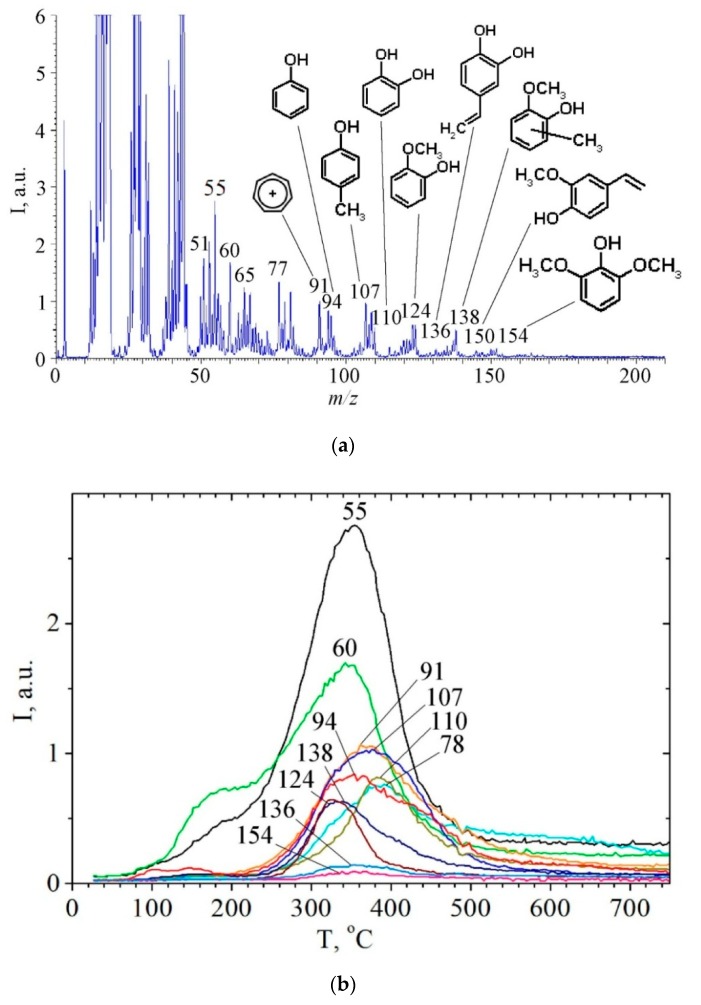
Mass spectra of pyrolysis products of rice husk hydrolytic lignin at 354 °C obtained after electron impact ionization (**a**), and TPD-curves of the ions with *m*/*z* 154, 138, 136, 124, 110, 107, 94, 91, 78, 60, 55 under pyrolysis of rice husk hydrolytic lignin (**b**).

**Figure 4 molecules-24-03075-f004:**
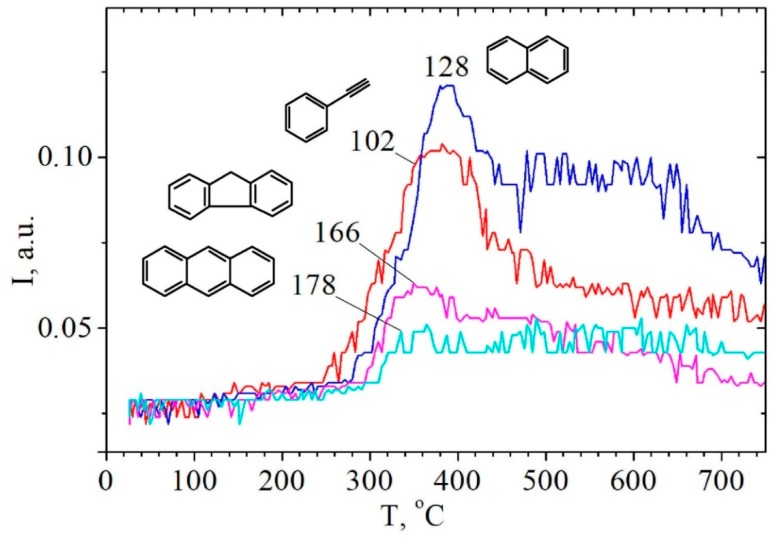
TPD-curves of the ions with *m*/*z* = 178, 166, 128, 102 obtained during pyrolysis of rice husk hydrolytic lignin.

**Figure 5 molecules-24-03075-f005:**
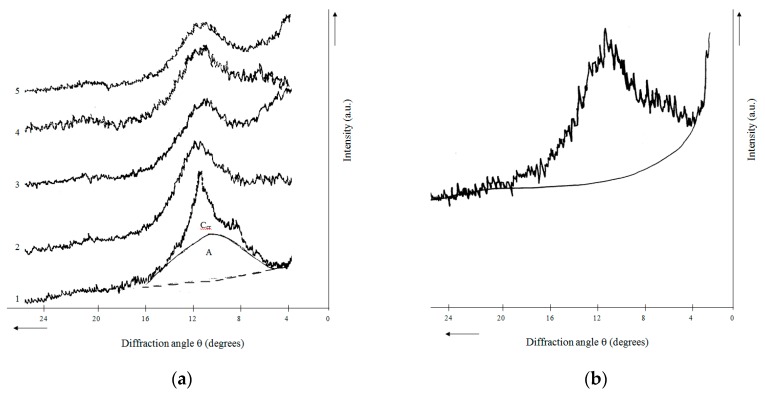
X-ray diffraction patterns of rice husk hydrolytic lignin and its carbonization products: (**a**) 1—L; 2—L-500; 3—L-650; 4—L-800; 5—L-1000; (**b**) Purified hydrolytic lignin.

**Figure 6 molecules-24-03075-f006:**
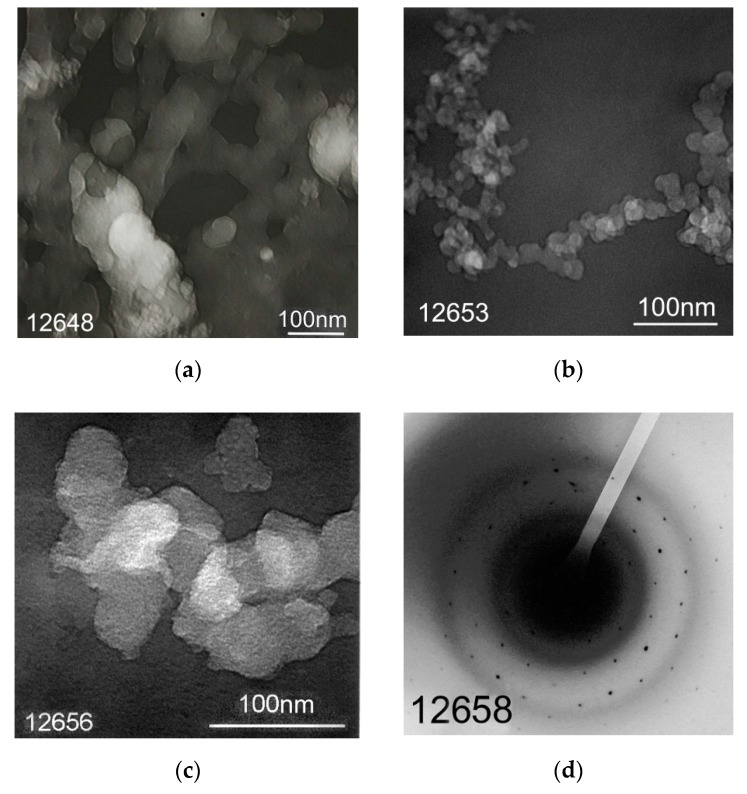
Morphology of rice husk hydrolytic lignin presented by different forms particles: (**a**) fibers; (**b**) dendritic formations; (**c**) rounded particles; (**d**) microdiffraction pattern.

**Figure 7 molecules-24-03075-f007:**
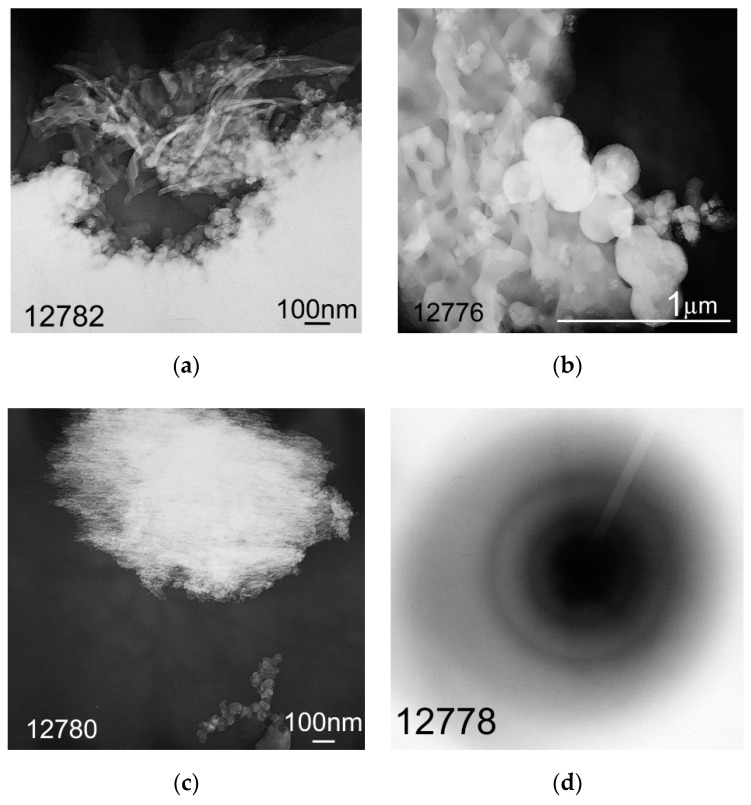
Morphology of rice husk hydrolytic lignin carbonization products presented by different forms particles: (**a**) fibers/tapes; (**b**) fibers and spherical particles; (**c**) isomorphic and dendritic formations; (**d**) microdiffraction pattern of carbon rounded particles.

**Figure 8 molecules-24-03075-f008:**
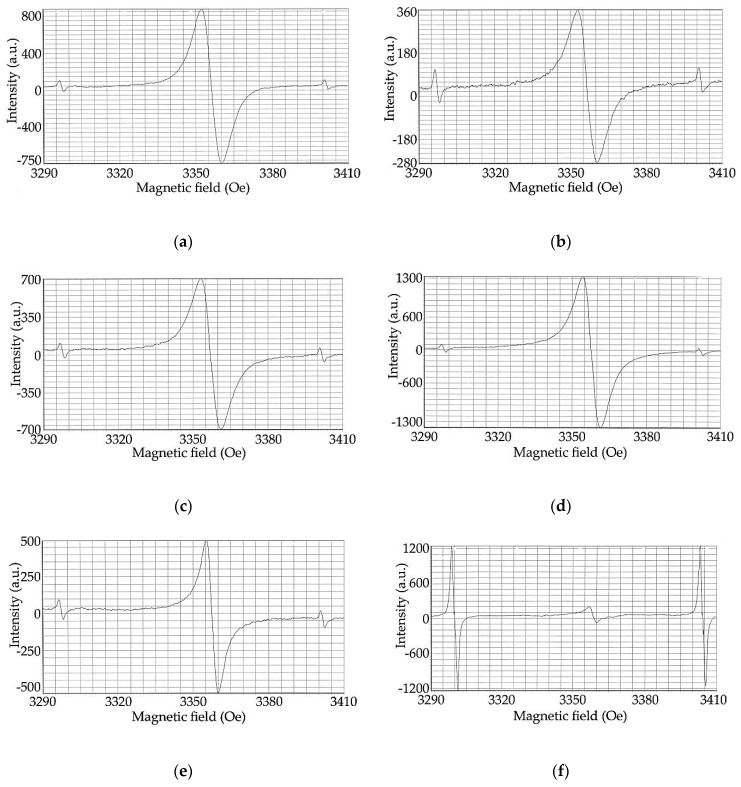
EPR spectra of rice husk hydrolytic lignin and its carbonization products: (**a**) L; (**b**) L-300; (**c**) L-450; (**d**) L-550; (**e**) L-650; (**f**) ions Mn^2+^ in MgO as reference sample.

**Table 1 molecules-24-03075-t001:** Elemental composition of hydrolytic lignin carbonization products.

Sample	Content, Mass %			
	C	H	N + O_org_	Mineral Residue
L-500 (OSG)	51.85	2.64	4.74	40.77
L-650 (OSG)	51.04	2.65	5.53	40.78
L-1000 (OSG)	49.80	2.65	5.79	41.76
L-500 (10–20 kPa)	44.31	2.04	3.65	50.00
L-650 (10–20 kPa)	42.2	2.00	3.20	52.60
L-1000 (10–20 kPa)	40,.54	1.36	3.00	55.10

In further studies, we used hydrolytic lignin carbonization products obtained in atmosphere of outgoing steam gas.

**Table 2 molecules-24-03075-t002:** Kinetic parameters (temperature of the maximum desorption rate *T_max_*, reaction order *n*, activation energy *E*^≠^, pre-exponential factor *ν*_0_, and activation entropy *dS*^≠^), temperature range (*T_range_*) of formation and peak intensities (*I*) of thermo-programmed reactions products during hydrolytic lignin pyrolysis.

Pyrolysis Product or its Fragment Ion	*m*/*z*^1^	*I*, a.u.	*T_range_*, °C	*T_max_*, °C	n	E ^1^, kJ/mol	*ν*_0_, s^−1^	dS ^1^, cal/(K × mol)	*R* ^2^ ^2^
Phenols									
Tropyliumion, C_7_H_7_^+^	91	1063	220–570	370	-	-	-	-	-
Phenol	94	0.840	220–600	355	-	-	-	-	-
Cresols	107	1035	240–530	370	-	-	-	-	-
Pyrocatechol	110	0.817	250–520	384	1	113	2.23 × 10^6^	−31	0.931
Guaiacol	124	0.633	250–500	340	-	-	-	-	-
Syringol	154	0.092	270–470	360	-	-	-	-	-
Methylguaiacol	138	0.646	250–430	330	1	142	1.02 × 10^10^	−14	0.947
4-Vinylphenols									
4-Vinylphenol	120	0.355	210–470	325	1	77	1.17 × 10^4^	−41	0.941
4-Vinylpyrocatechol	136	0.144	260–430	355	1	104	1.17 × 10^6^	−24	0.939
4-Vinylguaiacol	150	0.257	235–400	320	1	107	9.56 × 10^6^	−28	0.970
4-Vinyl-methylguaiacol	164	0.109	260–420	335	1	100	1.38 × 10^6^	−32	0.952
Aromatic/Polycyclic aromatic hydrocarbons									
Benzene	78	0.761	250–750…	390	-	-	-	-	-
Phenylacetylene	102	0.104	250–750…	380	-	-	-	-	-
Naphthalene	128	0.121	270–750…	390	-	-	-	-	-
9H-Fluorene	166	0.063	300–700	365	-	-	-	-	-
Anthracene or Phenanthrene	178	0.053	310–750…	-	-	-	-	-	-

^1^*m*/*z*—Ratio of ion mass to ion charge. ^2^*R*^2^—Coefficient of determination.

**Table 3 molecules-24-03075-t003:** X-ray phase composition and X-ray diffraction characteristics of graphite-like phase of rice husk hydrolytic lignin carbonization products.

Sample	Composition of Carbon-Containing Component, %			Parameters G		
	G	N	H	*d_002_*, nm	*L_c_*, nm	*L_a_*, nm
L-400	-	80	20	-	-	-
L-500	44	41	15	0.38	2.1	-
L-650	43	35	22	0.381	4.0	-
L-800	44	38	18	0.375	4.6	-
L-1000	62	-	38	0.375	4.7	-

**Table 4 molecules-24-03075-t004:** Parameters of EPR spectra of rice husk hydrolytic lignin and its carbonization products.

Sample	PMCs Concentration, spin g^−1^	EPR Line Width	g-Factor
L	7.7 · 10 ^16^	6.3	2.0032
L-300	3.6 · 10 ^16^	6.3	2.0030
L-450	6.5 · 10 ^16^	6.7	2.0030
L-550	3.1 · 10 ^17^	6.2	2.0028
L-650	1.6 · 10 ^17^	3.8	2.0027

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
