# Peer review of "Rice Husk Hydrolytic Lignin Transformation in Carbonization Process"

_molecules, 2019, doi:10.3390/molecules24173075_

Round 1

Reviewer 1 Report

The authors study lignin isolated from rice husk. The study is rudimentary, although well performed.

The TEM images need to be improved, and more discussion is needed about the fundamental mechanisms that support their final claim: "To conclude, structural transformations of rice husk hydrolytic lignin in the process of carbonization proceed through the stage of free radical formation followed by hexagonal networks formation of cyclically polymerized carbon."

Figure captions need to be better explained. For instance Figure 6 and 7 has 4 sections, and it is not explain what it is in each of them.

The final conclusion is too obvious: So, there is recommended to select conditions of rise husk hydrolytic lignin carbonization to produce material with required properties. Which conditions? which materials? Which properties?

Author Response

Dear Reviewer,

Authors would like to thank you for your reviewing and high estimate of our article. We understand it has been taken a lot of time to do this big job. Your comments are very important for us. We tried to pay attention to each one to improve the article. We hope we have done it. Please see our response attached.

As for English, we discuss item of English checking in MDPI English editing service.

Reviewer 2 Report

The authors present a manuscript entitled “Rice Husk Hydrolytic Lignin Transformation in Carbonization Process”. The topic of this manuscript falls within the scope of Molecules journal.

Following points need attention of authors during revising the manuscript.

                    The table 2, I did not find description of the m/z parameter

                    The methodology section is presented on the bottom of this manuscript.  What were the special reasons that authors shift the "Methods" section  to the end of the manuscript?

                    In “Introduction” section, authors need to explain how it is possible to thermal hydrolysis? Are the alternative methods to isolate the hydrolytic lignin (lignin) from fiber rich agri by-products, eg. without H2SO4?

                    Authors used some analysis eg. IR, EPR, there are innovative, but in the manuscript lack the precision information about methodology. Readers which would like to repeat your experiment will be little bit confused. Eg. Authors  in the EPR analysis section, have written that EPR spectroscopy analysis have been done at room temperature in optimal contitions for the spectra recording. Could Authors explain optimal conditions for the spectra recording?  I suggest to add the EPR profile line as a chart , to the manuscript. It will be helpful to describe of the EPR analysis.

                    Did the rice hulls were cleaned or sieved before use? Describe the rice hulls pre-treatment, please.

                    I suggest, the flowchart addition in the "Methods" section, for the clear description of the rice hulls treatment , step by step, during the experiment.

In conclusion. The reviewed manuscript presents very interesting and important research. But the structure of chapters is strange for me, Authors put in “Materials and methods” section on the end of the manuscript. Methods are described very shortly without important details. In my opinion the manuscript needs major revision.

Author Response

Dear Reviewer,

Authors would like to thank you for your reviewing and high estimate of our article. We understand it has been taken a lot of time to do this big job. Your comments are very important for us. We tried to pay attention to each one to improve the article. We hope we have done it. Please see our response attached.

Reviewer 3 Report

Although a relevant topic, the text needs several improvements (see the comments attached).

I suggest its revision according these comments and a new submission after that.

Author Response

Dear Reviewer,

Authors would like to thank you for your reviewing of our article. We understand it has been taken a lot of time to do this big job. Your comments are very important for us. We tried to pay attention to each one to improve the article. We hope we have done it. Please see our response attached.

As for English, we discuss item of English checking in MDPI English editing service.

Round 2

Reviewer 1 Report

They addressed my concerns, and the paper looks good to me.

Reviewer 2 Report

Thank you for provided changes. I recommend this manuscript for publication.